Manuscript in preparation for *Atmospheric Measurement Techniques*, EGU

# Method development and application for the analysis of chiral organic marker species in ice-cores

Johanna Schäfer[1], Anja Beschnitt[1], François Burgay[2,3], Thomas Singer[2,3,4], Margit Schwikowski[2,3,4], Thorsten Hoffmann[1]

[1]Department of Chemistry, Johannes Gutenberg University, Mainz, 55099, Germany
[2]Laboratory of Environmental Chemistry (LUC), Paul Scherrer Institute, Villigen PSI, 5232, Switzerland
[3]Oeschger Centre for Climate Change Research, University of Bern, 3012 Bern, Switzerland
[4]Department of Chemistry, Biochemistry and Pharmaceutical Sciences, University of Bern, 3012 Bern, Switzerland

*Correspondence to*: Thorsten Hoffmann (t.hoffmann@uni-mainz.de)

**Abstract.** Glaciers are valuable environmental archives that preserve organic compounds from atmospheric aerosols that can be used as marker species for their respective emission sources. Most environmental studies do not distinguish between the enantiomers of chiral compounds, although these compounds, mostly from biogenic sources, are very common in the atmosphere. We have developed a two-dimensional liquid chromatography (mLC-LC) method that allows the simultaneous determination of the chiral ratios of the monoterpene oxidation products cis-pinic acid and cis-pinonic acid in ice-core samples. The method combines a reversed-phase column in the first dimension and a chiral column in the second dimension in a simple instrumental setup with only one additional six-port valve. This novel method was successfully applied to selected ice-core samples from the Belukha glacier in the Siberian Altai spread over the period 1870-1970 CE. The chiral ratio of cis-pinic acid showed fluctuating values, while the chiral ratio of cis-pinonic acid remained more constant with an excess of the (–)-enantiomer.

## 1 Introduction

Volatile organic compounds (VOCs) are emitted by all plant species and thus enter the atmosphere in large quantities. The most common classes of these biogenic VOCs are isoprene and monoterpenes such as α- and β-pinene, which form low-volatile compounds after reaction with atmospheric ozone, OH and $NO_3$ radicals (Guenther et al., 1995; Kroll and Seinfeld, 2008). Gas-to-particle conversion of the reaction products leads to the formation of secondary organic aerosols (SOA), which then directly influence the Earth's radiation budget via absorption, reflection or scattering of incoming sunlight, or indirectly via their function as cloud condensation nuclei (Jokinen et al., 2015; Kourtchev et al., 2008). Many biogenic VOCs are chiral, i.e., they have stereoisomers whose structures are like mirror images of each other. In an achiral environment, these enantiomers share most physical and chemical properties, so in most studies no distinction is made in the analysis. However, in a stereogenic and therefore natural environment, chirality has a major impact on biological processes. Many organisms only release a certain enantiomer or at least a certain ratio of enantiomers, e.g., in the diverse interactions between plants and insects (Phillips et al., 2003; Mori, 2014; Malik et al., 2023) Prominent examples are the monoterpenes α- and β-pinene, each of which has an enantiomeric (+) and (-) form. The stereo information of chiral compounds can be retained in oxidation reactions and can therefore be found not only in the volatile precursors, but also in some of the low-volatile biogenic SOAs compounds. For example, the four-membered ring of the α- and β-pinene, which carries the stereo information, remains intact during their oxidation to cis-pinic acid or to cis-pinonic acid (Leppla et al., 2021; Ebben et al., 2011; Leppla et al., 2023). Interestingly, measurements at the ATTO tower (Amazonian Tall Tower Observatory) in Brazil have shown that the chiral ratio of cis-pinic acid does not always correlate with the chiral ratio of its main precursor, α-pinene. In this case it is assumed that the chiral ratio of the short-lived precursors (α- and β-pinene) depends more on local emissions, whereas the chiral ratios of the SOA components tend to reflect the larger-scale emissions (Zannoni et al., 2020; Leppla et al., 2021). Overall, there are only a few

enantioselective studies on pinene that have been carried out in ambient air. However, the measurements performed to date indicate a high degree of regiospecificity as well as temporal variations in the chiral composition of the measured VOCs (Williams et al., 2007; Staudt et al., 2019; Zannoni et al., 2020; Byron et al., 2022). The emission of pinene enantiomers strongly depends on the type of vegetation, even tree species. Yassaa et al. analyzed different chemotypes of Scots pine and spruce and were able to show that each of these different conifers emits different ratios of pinene enantiomers (Yassaa et al.,

2012). In spruce, for example, (-)-α-pinene predominates. The pinene composition in atmospheric aerosols is therefore strongly dependent on the local flora. In addition, different chiral signatures were found in the canopy, trunk and soil regions of a maritime pine forest (Staudt et al., 2019) and in the Brazilian rainforest (Zannoni et al., 2020). Byron et al. investigated the emissions of monoterpenes, including (+)-α-pinene, (-)-α-pinene and (-)-β-pinene, in a closed tropical rainforest ecosystem under extreme climatic conditions (Byron et al., 2022). Again, the enantiomers showed unique responses to drought and rain

phases, suggesting that enantiomer distribution is key to understanding the underlying processes driving monoterpene emissions from forest ecosystems and should be understood to predict atmospheric feedbacks in response to climate change.

Insights into past atmospheric conditions are essential for understanding recent environmental and climate change. A historical record of chiral monoterpenes could therefore shed light on the underlying dynamics of their emissions. Mid- and high-latitude

glaciers are particularly valuable environmental archives, because they are located close to the emission sources. Monoterpenes have short lifetimes of a few minutes to a few hours in the atmosphere (Kesselmeier et al., 2000) and have therefore not yet been measured directly in ice-cores. In contrast, the longer lifetimes of less reactive SOA compounds such as cis-pinic acid and cis-pinonic acid enable their atmospheric transport to glaciers, where they have already been successfully analyzed (Beschnitt et al., 2022; Müller-Tautges et al., 2016; Pokhrel et al., 2016; Fu et al., 2016; King et al., 2019b; King et al., 2019a;

Vogel et al., 2019), although no chiral distinction has been made so far. The aim of this study is therefore to develop a method to analyze the chiral distribution of these compounds in ice-core samples. Since the separation of the complex sample matrix with a simple chiral column is difficult due to its limited chemical selectivity, a multiple heart-cutting 2D-LC method (mLC-LC) was developed. Heartcut LC-LC is used to improve separation when individual analyte groups cannot be sufficiently resolved in a single dimension or when peak purity needs investigation. In this approach, a specific section of the flow from

the first dimension is selectively transferred to the second dimension ("heartcut"). This is achieved through an additional valve equipped with a sample loop whose volume is adequate to hold the aforementioned fraction of the first dimension eluent. When multiple regions are isolated in this way, the technique is referred to as multiple heartcut 2D-LC (mLC-LC). This method offers a substantial increase in resolution and selectivity for targeted compounds This technique is thus particularly suitable for the separation of enantiomers of chiral compounds, given that they coelute in the achiral first dimension due to their

identical physical and chemical properties. The use of a chiral column in the second dimension has been previously documented in the determination of enantiomeric ratios of D- and L-amino acids or in the assessment of pharmaceutical purity (León-González et al., 2014; Hildmann and Hoffmann, 2024; Pirok et al., 2019). The optimized method was applied to several samples of an ice-core originating from the Belukha glacier of the Siberian Altai in Central Asia.

## 2 Materials and methods

### 2.1 Materials

Ultrapure methanol (MeOH, LC/MS grade), water (LC/MS grade) and acetonitrile (ACN, LC/MS grade) were purchased from Fisher Scientific. Formic acid (99%, LC/MS) was obtained from VWR Chemicals. Ammonium hydroxide solution (analytical grade, 25%) was obtained from Honeywell Fluka. Hydrochloric acid (HCl, suprapure, 30%) was purchased from Merck KGaA. *Cis*-pinonic acid (99%)  as well as enantiomerically pure (–)- and (+)-α-pinene (99%, optical purity ee: 97%) were purchased

from Sigma-Aldrich. Ultrapure water with 18.2 MΩ cm  resistance was produced using a water purification system from Merck Millipore. A standard compound of *cis*-pinic acid was synthesized according to (Moglioni et al., 2000).

## 2.2 Sample preparation

The ice-core was recovered from the Belukha glacier (4062 m.a.s.l. - 49°48'27.7"N; 86°34'46.5"E ) by a team of the Paul Scherrer Institut (PSI, Villigen, Switzerland) together with scientist from the Institute of Water and Environmental Problems

(IWEP, Barnaul, Russia) in 2018 and was stored at –20 °C in a cold room at the PSI (Burgay et al., 2024). The diameter of the core was 7.8 cm. For analysis, 2 cm was removed from the outer layer to minimize potential contamination (Gambaro et al., 2008). The inner part was cut into sections, melted under helium-atmosphere and then filtered through a quarz-fiber filter. All the sections were taken from samples with density > 0.7 g/mL, indicating that the analyses were done exclusively on ice samples, and not firn. The analysed depths can be found in the Supplementary Material (Table S3).  Samples were transferred

to pre-cleaned amber glass vials with PTFE coated screw caps and stored at –25 °C until analysis. The cleaning procedure of the vials involved rinsing with ultrapure water and LC-MS grade methanol three times each and a bake-out at 450 °C for 8 hours (Bosle et al., 2014; Burgay et al., 2023). For analysis, the samples were melted at room temperature and aliquots of 10 mL were taken and set to pH 8 with 10 µL of a methanolic 5% ammonium hydroxide solution. The samples were loaded onto SPE cartridges (WAX, 3cc, 60 mg, Waters, Milford USA), which were preconditioned with methanol and 0.1% formic acid

in ultrapure water (6 mL each). The cartridges were then rinsed with 6 mL of ultrapure water and dried by blowing air through the cartridges. The compounds were eluted with three portions of 500 µL methanol, followed by three portions of 500 µL 5% methanolic ammonium hydroxide solution. The eluate was evaporated to dryness under a gentle stream of nitrogen at 30 °C and the residue was dissolved in 200 µL of $H_2O$/ACN (1:1, v/v) supported by sonification at 30 °C for 10 min. The samples were then filtered through PTFE filters (3mm, 0.2 µm, Altmann Analytik, Germany) and stored in the freezer at –25 °C.

Standard solutions of racemic cis-pinic acid and cis-pinonic acid were prepared in $H_2O$/ACN (1:1, v/v).

## 2.3 Chromatography and mass spectrometry

Measurements were performed using a UHPLC unit (Dionex Ultimate 3000, Thermo Fisher Scientific, Germany) coupled to a Q-Exactive Hybrid Quadrupole-Orbitrap mass spectrometer (Thermo Fisher Scientific, Germany). The ion source used was an ESI source operated in negative mode. The ESI probe was heated to 150 °C, and the capillary temperature was set to 320°C.

The sheath gas pressure was set to 60 psi, the auxiliary gas pressure to 20 psi, and the spray voltage to –3.5 kV. The mass spectrometer was operated in full-scan mode with a mass range of *m/z* 80–500 and a mass resolution of 70,000. The instrument was operated using XCalibur 4.3 software. Chromatographic separation of the analytes, cis-pinic acid and cis-pinonic acid, was performed using an Acquity UPLC CSH Fluoro-Phenyl (PFP) column (100x2.1 mm, 1.7 µm, Waters, USA). This corresponds to the first dimension of the 2D LC method and is referred to as such below. In the second dimension, chiral

separation of enantiomers was performed on an amylose-based tris(3-chloro-5-methylphenylcarbamate) column (150x2.1 mm ID, 5 µm, Daicel, CHIRALPAK IG). In both dimensions, separation was achieved with eluents A (water containing 2% ACN and 0.04% formic acid) and B (ACN containing 2% water) at a constant flow of 200 µL/min and a column oven temperature of 40 °C. The respective gradients are summarized in Table 1.

### 2.3.1 Two-dimensional separation

Two-dimensional separation was performed on the previously mentioned columns connected to two separate pumps. The target analytes were transferred from the first dimension (PFP) to the second dimension (chiral) via a second six-way valve and a 20 µL sample loop. Both pathways were combined prior to introduction into the mass spectrometer (Fig. 1), allowing for what can be described as simultaneous measurement of the first dimension's full chromatogram alongside the targeted separation of the enantiomers in the second dimension. The targeted enantiomers thus appear at higher retention times influenced by both

columns in an otherwise one dimensional separation. In position A, the sample loop is flushed with the eluents from pump 2 (second dimension), while the first dimensional separation is occurring. Upon the elution of the targeted analyte from the first dimension column, the valve is switched to position B, thereby filling the sample loop with the eluent from the first dimension, which contains the analyte peak. The valve is then switched back to position A, allowing the precut volume to be transferred to the chiral column in the second dimension via pump 2 and subsequently to the mass spectrometer.. The optimal time

windows for maximum analyte transfer proved to be at 2.70–2.80 min and 3.76–3.86 min for cis-pinic acid and cis-pinonic acid, respectively. The valve was therefore only in position B during these periods, otherwise position A was maintained.

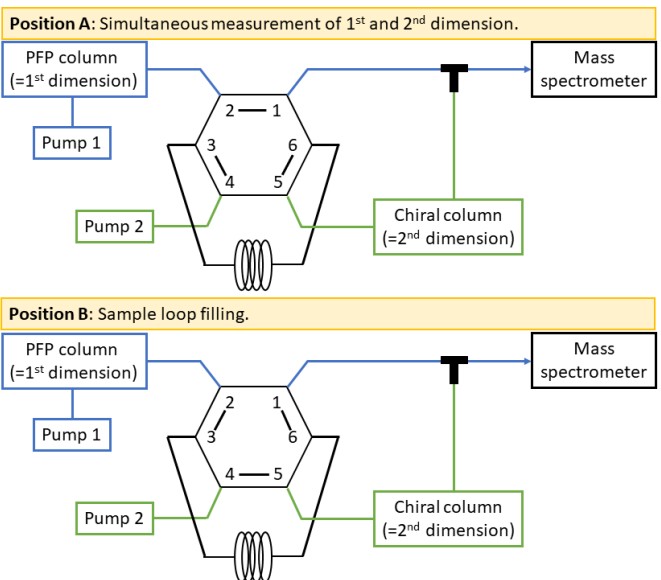

**Figure 1: Instrumental setup of the heartcut-2D-LC method. In the first dimension, separation is performed using a PFP column; in the second dimension, the target enantiomers are separated on a chiral column. In position A, the sample loop is rinsed with**
**solvent from pump 2. The heartcut is performed by switching to position B, whereby the target analyte from the first dimension is rinsed into the sample loop and then transferred to the chiral column by switching to A again. Both dimensions are combined via a T-piece and fed together into the mass spectrometer.**

**Table 1: Solvent conditions and valve positions of the 2D-LC method. Pump 1 operates the 1st dimension, pump 2 operates the 2nd dimension. Eluents A and C are 98% $H_2O$ with 2% ACN and 0.04% formic acid, eluents B and D are 98% ACN with 2% $H_2O$. Flow is 0.2 mL/min in both cases.**

| Pump 1 | | Pump 2 | | Valve | |
|---|---|---|---|---|---|
| Time (min) | A (%) | Time (min) | C (%) | Time (min) | Position |
| 0 | 80 | 0 | 85 | 0 | A |
| 15 | 80 | 8 | 85 | 2.7 | B |
| 15.5 | 1 | 8.1 | 80 | 2.8 | A |
| 17 | 1 | 15 | 80 | 3.76 | B |
| 17.5 | 80 | 15.5 | 1 | 3.86 | A |
| 18 | 80 | 16.6 | 1 | | |
| | | 17.5 | 85 | | |
| | | 18 | 85 | | |

## 2.4 Chamber experiments

The identification of the two enantiomers of cis-pinic acid and cis-pinonic acid requires enantiomerically pure standards. Since these are not commercially available, ozonolysis experiments were performed with enantiomerically pure (+)-α-pinene and (–)-α-pinene (Sigma Aldrich, 99%, optical purity 97% ee). The experiments were carried out in a 100 L glass reaction chamber,

which was darkened beforehand to exclude light-induced reactions. The chamber was connected to three gas inlets through which humidified air, ozone, and the volatile organic compound, in this case α-pinene, were introduced. For this purpose,

ambient air was drawn in through a compressor and organic compounds were removed from the air stream using an activated carbon trap. The dry air was fed into an internal ozone generator of an ozone analyzer (Dasibi, 1008-RS, USA), where ozone was generated by irradiation with UV light (flow rate 3 L/min). To generate humidified air, the airflow was passed through a gas wash bottle containing ultrapure water at a flow rate of 4 L/min. A vial of α-pinene was placed in a VOC test gas apparatus, which was operated at a constant temperature of 40 °C and a flow rate of 1 l/min (Thorenz et al., 2012). In addition, the

chamber was connected to a condensation particle counter (CPC, Porta Count Plus, TSI Corp., USA) to monitor the particle number and thus the progress of the ozonolysis reaction. Since a slight overpressure was to be maintained in the chamber to prevent ambient air from entering the chamber, pressure equalization was ensured via two wash bottles containing kerosene oil. In order to be able to collect the desired ozonolysis products on a filter, a pump was connected behind the filter holder and a rotameter and needle valve were connected in between in order to be able to regulate the gas flow. A flow rate of 5.5 l/min

was selected. The filters were made of borosilicate glass microfibers bonded with PTFE (Pallflex® Emfab, 70 mm diameter). Before conducting an experiment, the chamber was flushed overnight with humidified air to remove residual organic compounds from the chamber. The vial of enantiomerically pure α-pinene was weighed and then placed in the test gas source. The gas streams from the humidified air and the test gas source were adjusted to their respective values to introduce pinene into the chamber. After approximately two hours, the CPC and ozone generator were switched on. The particle count was

monitored and after about 90 minutes sampling was started by switching on the pump behind the filter holder. Depending on the run, sampling took between 2 and 4 hours. The VOC vial was then weighed again to determine the amount of pinene released from the test gas source. For analysis, the filters were cut into small pieces and extracted three times with 1.5 mL 9:1 $H_2O$/MeOH on a rotary shaker for 40 min each. The solutions were filtered through PTFE filters (3 mm, 0.2 μm, Altmann Analytik, Munich, Germany) and then evaporated under a gentle stream of nitrogen. After reconstitution in 9:1 ACN/$H_2O$,

they were sonicated at 30 °C for 10 min and stored in the freezer until measurement.

## 3 Results and discussion

### 3.1 Method development of heart-cut two-dimensional LC (m2D-LC)

Before using the two-dimensional approach, the separation of the analytes in each dimension was first optimised. The separation of cis-pinic acid enantiomers on the CHIRALPAK IG column is achieved with isocratic conditions (80:20) using

eluents A (water containing 2% ACN and 0.04% formic acid) and eluent B (ACN containing 2% water) (Machtejevas, 2021), as well as a flow rate of 200 μL/min and an oven temperature of 25 °C, as described in (Leppla et al., 2021). This method just about achieves baseline separation. However, poorer separation in the second dimension of the 2D LC method was expected because the analyte is less compactly transferred from the sample loop to the second dimension (Stoll and Carr, 2017). To improve the separation further, different eluent compositions, gradients, flow rates as well as oven temperatures were tested.

A summary of all conditions tested can be found in the Supplementary Material (Table S1, Table S2). A significant reduction in peak width was achieved by increasing the oven temperature to 40 °C and a slight alteration of the initial isocratic solvent composition to 85% eluent A. The optimized method was also tested with a cis-pinonic acid standard, and baseline separation was obtained. The final method thus includes isocratic conditions using eluents A (water containing 2% ACN and 0.04% formic acid) and B (ACN containing 2% water) with 85:15 until 8.0 min, followed by 80:20 at a constant flow of 200 μL/min

and a column oven temperature of 40 °C. In order to transfer cis-pinic acid and cis-pinonic acid sequentially to the chiral column in a mLC-LC approach, their retention times in the first dimension were determined and optimized. The retention times of the two analytes must differ such that it is possible to transfer the first peak to the sample loop and to rinse it completely before the next target analyte can be transferred. Both the size of the sample loop and the flow rates of both pumps were

considered. Since the CHIRALPAK IG column is operated in reverse phase-mode in the second dimension, the same eluents

A and B can be used in the first dimension for the PFP column. Sufficient separation of the analytes cis-pinic acid and cis-pinonic acid was achieved under the same isocratic conditions (80:20) used for the separation of the enantiomers in the second dimension. This eliminates all the disadvantages that can occur in 2D-LC due to solvent incompatibilities, such as peak broadening and poor resolution (Pirok et al., 2019). Three sample loop sizes, 100 µL, 50 µL and 20 µL, were tested for suitability. As the sample loop volume decreased, the separation of the enantiomers improved significantly. The 50 µL loop

already showed a significant improvement over the 100 µL loop, which only achieved minimal separation. The 20 µL sample loop then achieved baseline separation for both cis-pinic acid and cis-pinonic acid. Large sample loop sizes have the advantage of transferring a large amount of analyte, i.e. the entire analyte peak of the first dimension. If the sample loop volume is larger than the cut volume or the analyte peak in general, so-called "undersampling" can occur, where analytes that have been resolved in the first dimension experience remixing in the sample loop (Stoll and Carr, 2017). The sample loop also acts as the

injection volume of the second dimension, meaning the analyte cannot be compactly transferred to the column, because it is distributed over a large volume (for comparison, the injection volume of the first dimension is 10 µL). Small volumes allow a much more compact transfer of the analyte to the second dimension, but only a fraction of the peak from the first dimension can be transferred. It is therefore of great importance that the optimal time window is chosen to capture the maximum of the peak and thus the highest amount of analyte. In order to improve separation with the 50 µL sample loop and to counteract peak

broadening, higher flow rates were tested to flush the analytes as compactly as possible from the sample loop onto the chiral column. For cis-pinic acid, a significantly improved separation of the enantiomers was achieved at 0.4 mL/min, but baseline separation was still not obtained. For each tested sample loop volume and flow rate, the optimal time window for both heart-cuts was redetermined. Ultimately, the 20 µL sample loop was chosen because it provides baseline separation along with good detection limits. The final method conditions and valve positions are summarized in Table 1.

**3.2 Chamber experiments**

For comparison of the retention times, a standard solution containing cis-pinic acid and cis-pinonic acid was measured. As can be seen in Figure 2, this standard shows equal peak areas of the enantiomers in both cases after integration, indicating a racemic mixture. Ozonolysis of (+)-α-pinene led to the formation of the oxidation product with the shorter retention time, i.e., the left peak E1. Accordingly, the signal emerging at longer retention times, i.e. the right peak E2, can be assigned to the oxidation

products of (–)-α-pinene. Traces of the other enantiomer were detected in both cases, but this can be attributed to memory effects (Doussin et al., 2023). Traces of both cis-pinic acid and cis-pinonic acid remain attached to the chamber wall and are again evaporated and detected in subsequent experiments. Nevertheless, both enantiomers of cis-pinic acid and cis-pinonic acid E1 and E2 can be clearly assigned to (+)-α-pinene and (–)-α-pinene, respectively.

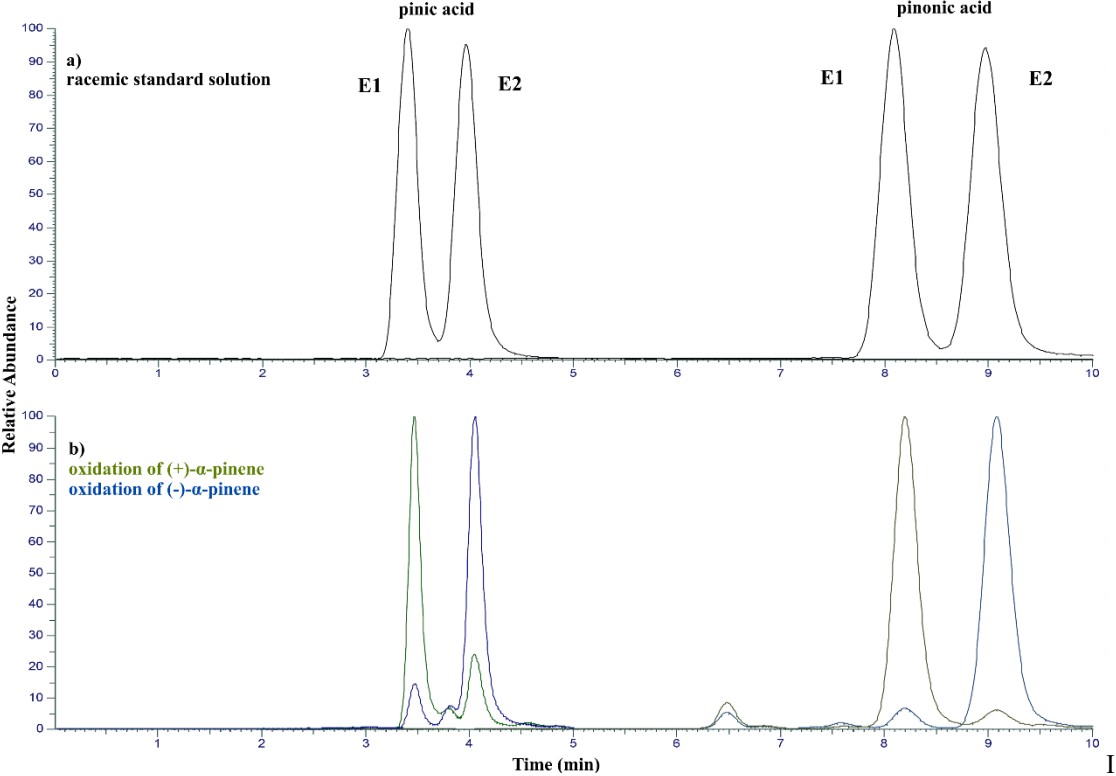

**Figure 2:** Overlay of the extracted ion chromatogram (XIC) of cis-pinic acid ([M–H]⁻ *m/z* 185.0819) and the XIC of cis-pinonic acid ([M–H]⁻ *m/z* 183.1026) of **a)** a racemic standard solution containing both enantiomers (E1 and E2) of the target compounds and **b)** the chamber experiment products with (+)-α-pinene (green) and (–)-α-pinene (blue) as precursors. For both cis-pinic acid and cis-pinonic acid, the enantiomer E1 (left peak) results from the ozonolysis of (+)-α-pinene, whereas the enantiomer E2 results from the ozonolysis of (–)-α-pinene. Note, that these measurements were not made using the developed m2D-LC method but with the chiral column alone, thus the retention times differ from those in Table 2.

### 3.3 Method validation

To validate the method, a calibration series of racemic standard solutions of cis-pinic acid and cis-pinonic acid and a blank sample of the composition 1:1 H₂O/ACN were measured. The instrumental limit of detection (LOD) was determined using the linear calibration method as the ratio of the threefold standard deviation of the blank and the respective slope of the calibration line. To assess the linearity, the regression coefficient $R^2$ of the calibration line was determined and the ratio of the signal intensity of the calibration standards and concentration S/c was plotted against the logarithmic concentration log(c). In addition to $R^2 > 0.999$ for all four enantiomers, good linearity was found for 10-100 ppb with a deviation of less than 5% from the mean value (Huber, 2007). The instrumental repeatability was calculated as the standard deviation of seven injections of a 100 ppb standard. Matrix-related effects were not considered, as potential negative influences, such as ion suppression in the ESI source, were deemed negligible due to the two-dimensional (2D) separation of the analytes, which decreases the chance of coelution. This separation is performed prior to analysis with an Orbitrap system, which is characterized by its high resolution and mass accuracy, thus further supporting the reliability of the method.

**Table 2:** Name, m/z value ([M–H]⁻), retention time in the final method $t_R$, instrumental limit of detection LOD and instrumental repeatability.

| Compound | m/z [M–H]⁻ | $t_R$ (min) | LOD (ppb) | Instr. repeatability (%RSD) |
|---|---|---|---|---|
| Cis-pinic acid E1 | 185.0819 | 6.55 | 0.14 | 3.1 |
| Cis-pinic acid E2 | 185.0819 | 7.29 | 0.29 | 3.2 |
| Cis-pinonic acid E2 | 183.1026 | 13.48 | 2.77 | 4.1 |
| Cis-pinonic acid E2 | 183.1026 | 14.17 | 2.11 | 2.4 |

### 3.4 Application of the method to selected ice-core samples

Figure 3 shows typical chromatograms of the final method applied to a racemic standard solution of cis-pinic acid and cis-pinonic acid and a sample of the Belukha ice-core. Both enantiomers of cis-pinic acid and cis-pinonic acid are present in the
ice-core sample. The amounts of all four enantiomers are clearly above the detection limit of the developed method and could be successfully separated from other components with identical *m/z* values. In some ice samples, other signals with *m/z* 185.0819 were observed, which partially overlap with the signal of the target enantiomer cis-pinic acid E2. This mass represents the deprotonated anion of $C_9H_{14}O_4$ and is typical for oxidation products of various other monoterpenes such as sabinic acid, 3-carenoic acid or limonic acid (from sabinene, Δ-3-carene and limonene) (Larsen et al., 2001; Kourtchev et al.,
2015). The identification of these signals is beyond the scope of this study. In the case of cis-pinonic acid, complete separation of both enantiomers was achieved without overlap with other signals, demonstrating that the mLC-LC approach is also suitable for highly complex samples.

The enantiomeric ratio E1/E2 of cis-pinic acid and cis-pinonic acid was determined for each of the seven selected Belukha
samples. These samples were selected because they showed baseline separation of the cis-pinic acid enantiomer E2 from other compounds with the same *m/z* value, which ensures an unambiguous determination of the chiral ratio. Figure 4 illustrates the ratios that were determined. E1 is the enantiomer that was formed by the oxidation of (+)-α-pinene, whereas E2 is the enantiomer that was formed by the oxidation of (-)-α-pinene. The diagram in Figure 4 demonstrates that the enantiomeric ratio of cis-pinic acid fluctuates significantly, within the range of 1.6 to 0.4.. In contrast, the enantiomeric ratio of cis-pinonic acid
is much more constant and fluctuates around a mean value of E1/E2 0.51 ± 0.16. Thus, very similar proportions of cis-pinonic acid enantiomers are present in all samples. The cis-pinonic acid ratio also shows that the concentration of the enantiomer E2 is approximately a factor of 2 higher than that of E1, indicating a temporally constant excess of the precursor (-)-α-pinene. The observation that the ratios of the two enantiomers for cis-pinic acid and cis-pinonic acid are different in the various samples may initially be surprising, since cis-pinic acid and cis-pinonic acid are associated in particular with the oxidation of α-pinene,
i.e., they are formed from the same precursor compound and should therefore contain the same stereogenic information. However, there are other variables that can influence the enantiomeric ratios of the oxidation products. For example, cis-pinic acid is also formed during the oxidation of ß-pinene, whereas this reaction does not lead to the formation of cis-pinonic acid (Glasius et al. 2000; Ma and Marston 2008; Larsen et al. 2001). This is simply due to the fact that the methyl ketone group of cis-pinonic acid cannot originate from the exocyclic methylene group of ß-pinene, whereas the corresponding further oxidized
carboxyl group of cis-pinic acid can. The E1/E2 cis-pinic acid ratios in Figure 4 can therefore also contain contributions from the ß-pinene oxidation, which can change the enantiomeric ratio of cis-pinic acid, but not that of cis-pinonic acid. However, due to a lack of information on both the ß-pinene precursor concentrations in comparison to the α-pinene concentration and the corresponding ratios of the precursor enantiomers, no further statements can be made.

In principle, the particular conditions prevailing at the time of formation during the oxidation of the biogenic VOCs can also contribute to changes in the E1/E2 ratios, simply because the oxidation of both α- and ß-pinene is initiated by both ozone and OH radicals, whose reaction pathways can lead to different product yields in relation to the measured organic acids. The loss pathways of α-pinene due to reactions with OH and $O_3$ are comparable, while under typical daytime conditions the OH oxidation of ß-pinene is an order of magnitude faster than ozonolysis (Lee et al., 2023). From this point of view, conclusions
about the respective oxidation regimes at the time of BVOC oxidation would of course be extremely interesting, but cannot be drawn with the present data set due to the insufficiently determinable boundary conditions (especially with regard to the chiral ratios of the precursor compounds).

The data shown here are the first reporting the enantiomeric ratios of SOA components from monoterpenes in ice-cores and thus on a chronological scale extending back in time. Unfortunately, there are very few enantioselective studies on pinenes in general. However, measurements have been carried out in the atmosphere above tropical forests in South America (Williams et al., 2007; Leppla et al., 2021; Zannoni et al., 2020) or boreal forests in northern Europe (Williams et al., 2007; Yassaa et al., 2012). In these studies, (+)- and (-)-α-pinene as well as (+)- and (-)-β-pinene were measured in ambient air. Leppla and coworkers have determined the concentrations of cis-pinic acid enantiomers in aerosol particles in the tropics (Leppla et al. 2023). Interestingly, the predominant α-pinene species found in boreal forests is the (+)-enantiomer (Williams et al., 2007), in contrast to the predominant (-)-enantiomer found in this work. Since the vegetation in the Altai region is dominated by boreal forests, it was expected that similar results would be obtained. These results are more in line with those observed for tropical forests (Williams et al., 2007). Here, both (-)-α-pinene and (+)-β-pinene dominate. In all studies, the enantiomeric ratios of α-pinene and β-pinene differ from each other. However, it should be noted that the emission of pinene enantiomers strongly depends on the local flora, even down to specific tree species (Yassaa et al., 2012). The varying ratios found in this study might therefore reflect changes in vegetation over time. These studies make evident, on the one hand, that information on the respective enantiomers has the potential to provide important insights into the respective climatic conditions and, on the other hand, how little research has been done to analyze chiral monoterpenes and their chiral oxidation products in general. Historical records of chiral monoterpene oxidation products, together with complementary information such as pollen abundance, could therefore shed light on the underlying dynamics of their emissions and possibly the environmental conditions prevailing during their formation.

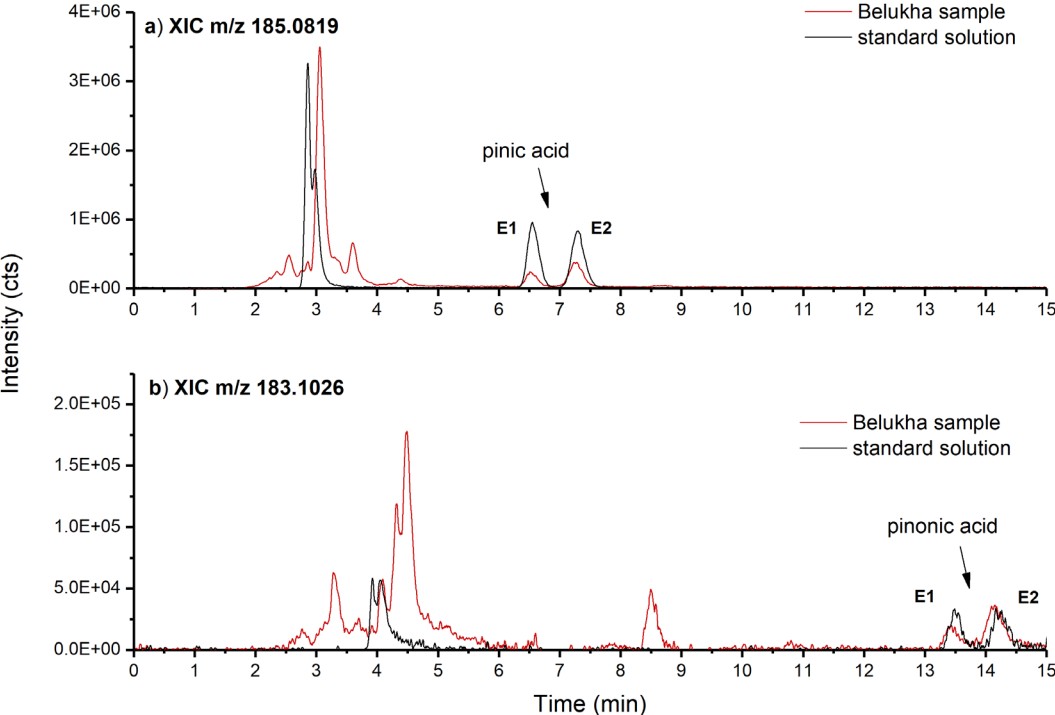

**Figure 3:** Typical chromatograms of the final method applied to a racemic standard solution of cis-pinic acid and cis-pinonic acid (black) and a Belukha sample (red). **a)** Extracted ion chromatogram (XIC) of m/z 185.0819 (cis-pinic acid) and **b)** XIC of m/z 183.1026 (cis-pinonic acid).

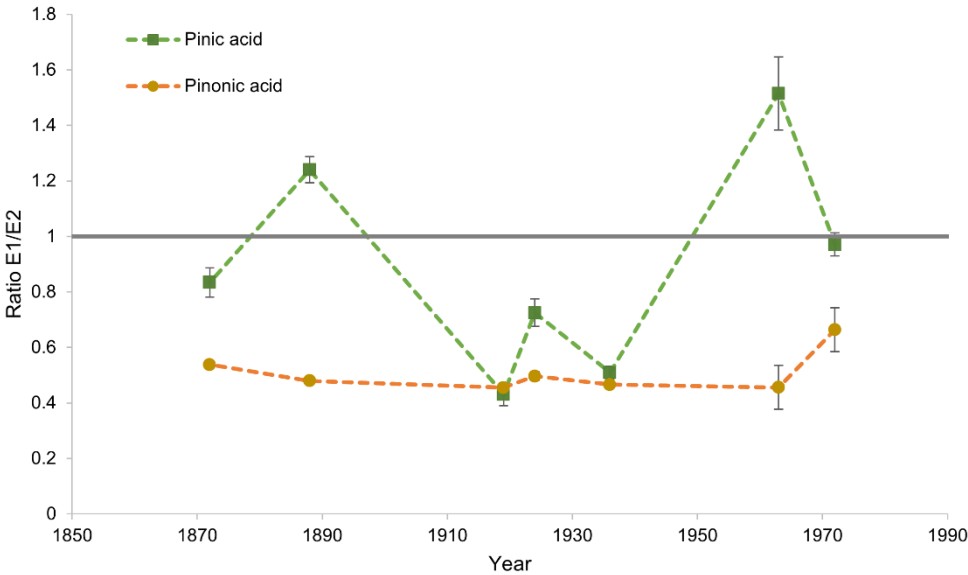

**Figure 4:** Chiral ratio E1/E2 cis-pinic acid (green) and cis-pinonic acid (yellow) in Belukha ice-core. In the previous chamber experiment it was proven that the enantiomer E1 of cis-pinic acid and cis-pinonic acid is the oxidation product of (+)-α-pinene and E2 of (–)-α-pinene (Figure 2). The chiral ratio of cis-pinonic acid remains relatively constant over the period under review with an excess of E2. The chiral ratio of cis-pinic acid strongly fluctuates over time. Error bars correspond to one standard deviation of twofold measurements. Note that some error bars are smaller than the shown data point. The grey line at E1/E2=1 indicates a racemic mixture. The dotted line through the data points is intended to lead the eye.

## 4. Conclusions

In this study, an m2D-LC method for the simultaneous determination of the enantiomeric ratios of chiral compounds in ice-cores is presented. Cis-pinic acid and cis-pinonic acid are some of the most prominent oxidation products of α- and ß-pinene formed in the atmosphere by ozonolysis and OH oxidation and are therefore of great interest as specific marker compounds for natural secondary aerosol formation processes. First, the method was developed for each dimension, i.e. for the PFP column and the chiral column. Subsequently, a 2D LC setup was chosen in which the flows of the first dimension (PFP) and the second dimension (chiral) were combined via a T-piece before being detected by the mass spectrometer. For two-dimensional chromatography, the influence of the sample loop volume and the injection rates was investigated to determine the optimum time window for the transfer to the second dimension. The smaller the sample loop volume and the higher the wash-in rate, the better the separation on the chiral column, as the analyte can be transferred compactly and less diluted. Using a 20 µL sample loop, baseline separation was achieved for the enantiomers of both analytes in the second dimension. Furthermore, the enantiomeric signals of cis-pinic acid and cis-pinonic acid were successfully assigned to their pinene precursors by performing ozonolysis chamber experiments with enantiomerically pure (+)- and (-)-α-pinene. The optimized method was applied to selected Belukha ice-core samples as a proof of concept, and the enantiomeric ratios of cis-pinic acid and cis-pinonic acid were successfully determined. The ratio of cis-pinic acid showed fluctuating values over time, while the ratio of cis-pinonic acid with an excess of the (-)-enantiomer remained rather constant over the observed period. Such historical records of the chiral composition of biogenic SOA markers from ice-cores can provide additional information on changing environmental conditions together with other proxies (pollens, major ions or other organics tracers), and thus expand our knowledge on past environmental conditions, e.g., the occurrence of droughts or different vegetation types.

## Data availability

The data in the study are available upon request (t.hoffmann@uni-mainz.de).

**Author contributions**

JS, AB, MS and TH designed the research. JS and AB carried out the measurements. JS led the writing, with significant input from TH as well as further input from all other authors.

**Competing interests**

The authors declare that they have no conflict of interest.

**Acknowledgments**

The authors thank the Deutsche Forschungsgemeinschaft (DFG, Bonn Germany) for financial support (HO 1748/20-1 Project number: 421860192).

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
