# Peer review of "Method development and application for the analysis of chiral organic marker species in ice-cores"

_EGUsphere, 2024_

## Author Comment (AC2)

**Method development and application for the analysis of chiral organic marker species in ice cores – Supplementary information**

Johanna Schäfer[1], Anja Beschnitt[1], Francois Burgay[2], Thomas Singer[2], Margit Schwikowski[2], Thorsten Hoffmann[1]

[1]Department of Chemistry, Johannes Gutenberg University, Mainz, 55099, Germany
[2]Laboratory of Environmental Chemistry (LUC), Paul Scherrer Institute, Villigen PSI, 5232, Switzerland

*Correspondence to*: Thorsten Hoffmann (t.hoffmann@uni-mainz.de)

**Figure S1. Schematic experimental setup of the chamber experiments with α-pinene.**

**Table S1. Tested conditions during 1D-LC method development with the chiral column, with tested eluents, column oven temperature T, flowrate and flowrate of the PCF.**

| | Eluent A (%) | Eluent B (%) | $T$ (°C) | Flowrate ($\mu$L/min) | Flowrate PCF ($\mu$L/min) |
|---|---|---|---|---|---|
| Eluent | 98% $H_2O$, 20% ACN, 0.04% formic acid | 98% ACN, 2% $H_2O$ | | | 50mM $NH_4OH$ in MeOH |
| 1 | 80 | 20 | 25 | 200 | |
| 2 | 90 | 10 | 25 | 200 | |
| 3 | 85 | 15 | 25 | 200 | |
| 4 | 85 | 15 | 30 | 200 | |
| 5 | 75 | 25 | 30 | 200 | |
| 6 | 80 | 20 | 30 | 150 | |
| 7 | 80 | 20 | 30 | 250 | |
| 8 | 75–>90 | 25–>10 | 30 | 200 | |
| 9 | 75–>80 | 25–>20 | 30 | 200 | |
| 10 | 80 | 20 | 40 | 200 | |
| Eluent | MeOH | ACN | | | |
| 1 | 10 | 90 | 30 | 200 | |
| 2 | 5 | 95 | 30 | 200 | |
| 3 | 2 | 98 | 30 | 200 | |
| 4 | 20 | 80 | 30 | 200 | |
| 5 | 50 | 50 | 30 | 200 | |
| 6 | 10 | 90 | 30 | 200 | 100 |
| Eluent | IPA | ACN | | | |
| 1 | 10 | 90 | 30 | 200 | 100 |
| 2 | 20 | 80 | 30 | 200 | 100 |
| 3 | 5 | 95 | 30 | 200 | 100 |
| 4 | 2 | 98 | 30 | 200 | 100 |
| 5 | 40 | 60 | 30 | 200 | 100 |

**Table S2. Tested conditions during 2D-LC method development with the PFP (pump 1) and chiral column (pump 2), with eluent composition and flowrate of each pump, column oven temperature T, flow rate of the PCF and the tested time windows for peak transfer.**

| Sample loop volume (µL) | Pump 1 (PFP) B (%) | Pump 1 (PFP) Flowrate (µL/min) | Pump 2 (chiral) B (%) | Pump 2 (chiral) Flowrate (µL/min) | Temp (°C) | Pin t(cut) (min) | Pinon t(cut) (min) | PCF (µL/min) |
|---|---|---|---|---|---|---|---|---|
| 100 | 20 | 300 | 20 | 200 | 40 | 1.75-2.00 | 2.55-2.80 | |
| 100 | 20 | 300 | 20 | 200 | 40 | 1.55-1.80 | 2.35-2.60 | |
| 100 | 20 | 300 | 20 | 200 | 40 | 1.45-1.70 | 2.35-2.60 | |
| 100 | 20 | 300 | 20 | 200 | 40 | 1.65-1.90 | 2.40-2.65 | |
| 100 | 20 | 300 | 20 | 300 while rinsing | 40 | 1.65-1.90 | 2.40-2.65 | |
| 100 | 20 | 300 | 20 | 400 while rinsing | 40 | 1.65-1.90 | 2.40-2.65 | |
| 50 | 20 | 300 | 20 | 200 | 40 | 1.70-1.85 | 2.45-2.60 | |
| 50 | 20 | 300 | 20 | 200 | 40 | 1.75-1.90 | 2.40-2.55 | |
| 50 | 20 | 300 | 20 | 200 | 40 | 1.725-1.875 | 2.425-2.575 | |
| 50 | 20 | 300 | 20 | 200
300 while rinsing | 40 | 1.70-1.85 | 2.40-2.55 | |
| 50 | 20 | 300 | 20 | 300
400 while rinsing | 40 | 1.70-1.85 | 2.40-2.55 | |
| 50 | 20 | 300 | 10 | 200 | 40 | 1.70-1.85 | 2.40-2.55 | |
| 50 | 20 | 300 | 10 | 400
200 at 5.5min | 40 | 1.70-1.85 | 2.40-2.55 | |
| 50 | 20 | 300 | 15 | 350
200 at 5.5min | 40 | 1.70-1.85 | 2.40-2.55 | |
| 50 | 20 | 300 | 20 | 200 | 40 | 1.70-1.85 | 2.40-2.55 | 100 |
| 50 | 20 | 300 | 10 | 400
200 at 5.5min | 40 | 1.70-1.85 | 2.40-2.55 | 100 |
| 20 | 20 | 200 | 20 | 200 | 40 | 2.745-2.845 | 3.775-3.875 | |
| 20 | 20 | 200 | 20 | 200 | 40 | 2.75-2.85 | 3.75-3.85 | |
| 20 | 20 | 200 | 15 | 200 | 40 | 2.80-2.90 | 3.75-3.85 | |
| 20 | 20 | 200 | 15 | 200 | 40 | 2.65-2.75 | 3.75-3.85 | |
| 20 | 20 | 200 | 15 | 200 | 40 | 2.725-2.825 | 3.75-3.85 | |
| 20 | 20 | 200 | 15 | 200 | 40 | 2.70-2.80 | 2.77-3.87 | |
| 20 | 20 | 200 | 15 | 200 | 40 | 2.70-2.80 | 3.76-3.86 | |

**Table S3. Dating and depth from the top of the ice core of the analysed Belukha ice core samples.**

| Depth (cm) | Age |
|---|---|
| 3402.77 | 1962 |
| 3756.92 | 1953 |
| 4451.95 | 1936 |
| 4795.83 | 1927 |
| 5326.95 | 1912 |
| 6030.70 | 1888 |
| 6418.82 | 1873 |

**Table S4. Accuracy (Acc.) of different concentrations (25 ppb, 50 ppb and 250 ppb) as percent relative deviation with respect to the standard concentration.**

| Compound | Acc.(25 ppb) (%) | Acc.(50 ppb) (%) | Acc.(250 ppb) (%) |
|---|---|---|---|
| Pinic acid E1 | 3.4 | 7.2 | 4.6 |
| Pinic acid E2 | 5.6 | 7.4 | 5.4 |
| Pinonic acid E2 | 14.0 | 6.4 | 4.1 |
| Pinonic acid E2 | 9.6 | 7.1 | 5.3 |

---

## Author Response (AR1)

**RC 1**

This work presents a two-dimensional liquid chromatography (mLC-LC) method for the simultaneous determination of the chiral ratios of monoterpene oxidation products in ice-core samples. The method was applied to ice-core samples from the Belukha glacier, revealing fluctuating chiral ratios for *cis*-pinic acid and more stable ratios for cis-pinonic acid. The findings have the potential to impact related fields (e.g., aerosol chemistry), extending beyond the immediate scope of the ice-core analysis. While the work is both important and novel, some sections could be improved to enhance its quality.

We thank the referee for the supportive review and the valuable and constructive comments/suggestions that helped to improve our manuscript. We have carefully revised the manuscript accordingly. Below you will find our point-by-point responses. Reviewer comments and suggestions are written in black, responses in blue. Changes in the manuscript are marked with "".

The introduction is well-written and informative. However, since the main emphasis of the paper (as indicated by the title) is on method development, it would be helpful to include some rationale and an overview of the existing literature (at least a couple of lines) on the application of multiple heart-cutting 2D-LC. Why was this technique chosen over others? Some explanation on what "heart-cutting" 2D-LC should be provided as it is not a commonly used technique especially for AMT readers.

We would like to thank the referee for this comment, we agree that an overview of 2D-LC applications would greatly improve the introduction and benefit any reader who is unfamiliar with the technique. We added the following text to clarify why we chose 2D-LC and explained the related term 'heart-cutting'.

Line 63: "Heartcut LC-LC is used to improve separation when individual analyte groups cannot be sufficiently resolved in a single dimension or when peak purity needs investigation. In this approach, a specific section of the flow from the first dimension is selectively transferred to the second dimension ("heartcut"). This is achieved through an additional valve equipped with a sample loop whose volume is adequate to hold the aforementioned fraction of the first dimension eluent. When multiple regions are isolated in this way, the technique is referred to as multiple heartcut 2D-LC (mLC-LC). This method offers a substantial increase in resolution and selectivity for targeted compounds This technique is thus particularly suitable for the separation of enantiomers of chiral compounds, given that they coelute in the achiral first dimension due to their identical physical and chemical properties. The use of a chiral column in the second dimension has been previously documented in the determination of enantiomeric ratios of D- and L-amino acids or in the assessment of pharmaceutical purity (León-González et al. 2014; Hildmann und Hoffmann 2024; Pirok et al. 2019)."

The authors use water containing 2% ACN and 0.04% formic acid (A) and ACN containing 2% water (B). It would be helpful to provide a rationale for adding 2% ACN to phase A and 2% water to phase B, and /or citing appropriate literature (considering the journal's wider readership and the paper's focus on method development).

A small amount of water is added to eluent B and acetonitrile to eluent A to improve mixing efficiency by bringing the two mobile phases closer in viscosity and surface tension. The addition of formic acid suppresses ionization of slightly acidic compounds, as the protonated and deprotonated

forms have different retention on the separation columns, which can lead to peak broadening. We added an appropriate reference.

Line 179: Ref. "(Machtejevas 2021)"

Line 176: The term "cut" in the sentence: "…and to rinse it completely before making the next cut" is unclear. Please rephrase for clarity.

We thank the referee for this remark and rephrased the term to convey the information more clearly. The 'next cut' refers to the next target analyte that should be transferred from the first dimension to the second one and should not be mixed with the previous contents of the sample loop.

Line 192: "The retention times of the two analytes must differ such that it is possible to transfer the first peak to the sample loop and to rinse it completely before the next target analyte can be transferred."

Line 185: Replace "rudimentary" with "minimal."

The text has been adapted accordingly.

Line 199: "The 50 µL loop already showed a significant improvement over the 100 µL loop, which only achieved minimal separation."

Section 3.3: Please include information on quality controls (QCs) and system suitability assessments.

We thank the reviewer for their help in improving our manuscript with regard to method validation and address the related comments below.

Please clarify the number of injections in the following statement: "The instrumental repeatability was calculated as the standard deviation of multiple injections of a 100 ppb standard."

The instrumental repeatability was calculated from seven injections of the 100 ppb standard. We corrected the referenced sentence in the manuscript accordingly.

Line 237: "The instrumental repeatability was calculated as the standard deviation of seven injections of a 100 ppb standard."

Why was the assessment done at 100 ppb (the upper limit of the reported linear dynamic range)? Typically, this is performed at low, medium, and high concentration levels (but below the max range, e.g. 70%).

The 100 ppb standard was initially selected to achieve signals of higher intensity while still maintaining the aforementioned linearity range. The lower concentrations were not measured sevenfold but threefold. The instrumental repeatability of these lower levels is comparable with that of the 100 ppb standard, with 10 ppb being 1.5-2.5% and 50 ppb being 1.9-3.5% across all four enantiomers.

Line 249: Add "whereas" before "E2 is formed by the oxidation of (-)-α-pinene."

The text has been adapted accordingly (Line 261).

Line 250: Clarify what is meant by "the diagram" and make an appropriate reference. I assume the authors are referring to Figure 4 (as mentioned in the previous sentence).

We rephrased the sentences with a proper reference to Figure 4.

Line 260: "Figure 4 illustrates the ratios that were determined. E1 is the enantiomer that was formed by the oxidation of (+)-α-pinene, whereas E2 is the enantiomer that was formed by the oxidation of (-)-α-pinene. The diagram in Figure 4 demonstrates that the enantiomeric ratio of cis-pinic acid fluctuates significantly, within the range of 1.6 to 0.4.."

Matrix effect is an important part of analytical method validation. Could the authors provide data or explain why this was not addressed in their study? While I realise that the authors rely on the high resolution and mass accuracy of the Orbitrap system, matrix effects, such as ion suppression, could be significant when using ESI, which is known to suffer from competitive ionisation. Was matrix-matched calibration used? This should be clarified in the validation section and addressed in the method application section as well.

As noted by the reviewer, we are indeed leveraging the high resolution and mass accuracy of Orbitrap MS. This is further supported by a reduced chance of coelution, thanks to twofold (2D) separation, which decreases competitive ionization effects, therefore deeming such matrix effects negligible. This study focuses exclusively on the instrumental method; sample preparation will not be addressed, and therefore, matrix effects specific to ice cores are not considered in detail. We also aim to extend the application of this method to additional sample types, such as atmospheric secondary organic aerosols (SOAs) collected on filters, and thus did not use matrix-matched calibration in the scope of this work. We will refrain from including it in the application section due it's negligibility and to avoid repetition but will include the following in the method validation section:

Line 238: "Matrix-related effects were not considered, as potential negative influences, such as ion suppression in the ESI source, were deemed negligible due to the two-dimensional (2D) separation of the analytes, which decreases the chance of coelution. This separation is performed prior to analysis with an Orbitrap system, which is characterized by its high resolution and mass accuracy, thus further supporting the reliability of the method."

What are the accuracy and precision (at a minimum of three concentration levels to ensure reliability across the analytical range. These levels generally include: low concentration (near the limit of quantification, or LOQ), medium concentration (close to the middle of the expected range) and high concentration (near the upper limit of the calibration curve). Again, considering the scope, the title of the paper and the type of samples, this is the minimum requirement in the method validation process to ensure that the analytical technique performs reliably across the specified concentration range.

We would like to thank the reviewer for their helpful advice, we added accuracy calculations to the Supplementary Material (Table S4).

Instrumental repeatability was evaluated through seven injections of the same 100 ppb standard solution conducted across two different analysis days, serving as a substitute for precision. While we recognize that precision encompasses broader assessments, such as testing the method in different laboratories, these additional measurements were beyond the scope of this work.

Lastly, I suggest using absolute, rather than relative, intensity in Figure 3 to better demonstrate the true fluctuations in signal. This would provide a clearer picture of method robustness and any potential interferences.

We are grateful to the author for their valuable suggestion. In response, we have replaced Figure 3 and have opted to display the absolute intensity in the manuscript, rather than the relative intensity.

[Figure]

**RC 2**

This is a pretty straightforward paper. It describes an analytical methodology suitable for separating the two enantiomers (mirror image molecules) of a couple of organic compounds present in mid-latitude ice cores. A few values are shown as proof of concept. The method seems clever, simple and sound, as well as novel. The paper is written at a technical level that would be more suitable for an analytical chemistry journal, and I think a little extra explanation might be needed for a journal that is read by practitioners without that knowledge. I think that, in a journal that tries to link measurements to applications, it would be appropriate to write a little more about the motivation for this type of analysis, especially in the light of the results found (I will expand on this below, final comment).

We thank the referee for the supportive review and the valuable and constructive comments/suggestions that helped to improve our manuscript. We have carefully revised the manuscript accordingly. Below you will find our point-by-point responses. Reviewer comments and suggestions are written in black, responses in blue. Changes in the manuscript are marked with "".

Specific comments:

Line 72 "Enantiomerically pure (–)- and (+)-α-pinene (99%, optical purity ee: 97%) as well as cis-pinonic acid". Please clarify, is the cis-pinonic also purchased as two separate enantiomers. I think not but from this sentence I am left unsure.

The reviewer is right in their assumption, the *cis*-pinonic acid was purchased as a racemic mixture containing both enenatiomers. We rephrased the sentence to make this more intuitive to the reader.

Line 81: *"Cis*-pinonic acid (99%) as well as enantiomerically pure (–)- and (+)-α-pinene (99%, optical purity ee: 97%) were purchased from Sigma-Aldrich."

Section 2.2. Please give a little more detail about the ice core. What diameter was the core (10 cm?)? What depths were analysed, and in particular are the sections firn or solid ice (this is very important for the likelihood of contamination penetrating the sample, much more likely in firn).

We are happy to provide more information about the ice core. We have added the following to the manuscript:

Line 87: "The diameter of the core was 7.8 cm. [...] All the sections were taken from samples with density > 0.7 g/mL, indicating that the analyses were done exclusively on ice samples, and not firn. The analysed depths can be found in the Supplementary Material (Table S3). "

Line 79-80. "The outer section was removed". Please be more precise: "x mm was removed from the outside". Did you do any tests to assure yourself that this was enough, eg by seeing if concentrations vary with distance from the edge of the sample?

The outer 2 cm were removed from the outside. The manuscript was adapted accordingly (Line 89).

No test was done to see if concentration varies with distance. However, previous studies performed on other organic tracers indicate that removing at least 2 cm of the outer section is enough to ensure

a representative characterization of the sample, without any possible contamination due to sample handling (see Figure 3 in Gambaro et al., 2008).

Gambaro, Andrea; Zangrando, Roberta; Gabrielli, Paolo; Barbante, Carlo; Cescon, Paolo (2008): Direct determination of levoglucosan at the picogram per milliliter level in Antarctic ice by high-performance liquid chromatography/electrospray ionization triple quadrupole mass spectrometry. In: Analytical Chemistry 80 (5), S. 1649–1655. DOI: 10.1021/ac701655x.

Line 62 and elsewhere. Please explain at least once what the term "heart-cut" means as this will be unfamiliar to most readers, even to analytical chemists I think.

We appreciate the referee for highlighting this point, which aligns with a comment from another referee. In response, we have added a section to the introduction that addresses heart-cut two-dimensional liquid chromatography (2DLC), including an explanation of the term "heart-cut."

Line 63: "Heartcut LC-LC is used to improve separation when individual analyte groups cannot be sufficiently resolved in a single dimension or when peak purity needs investigation. In this approach, a specific section of the flow from the first dimension is selectively transferred to the second dimension ("heartcut"). This is achieved through an additional valve equipped with a sample loop whose volume is adequate to hold the aforementioned fraction of the first dimension eluent. When multiple regions are isolated in this way, the technique is referred to as multiple heartcut 2D-LC (mLC-LC). This method offers a substantial increase in resolution and selectivity for targeted compounds This technique is thus particularly suitable for the separation of enantiomers of chiral compounds, given that they coelute in the achiral first dimension due to their identical physical and chemical properties. The use of a chiral column in the second dimension has been previously documented in the determination of enantiomeric ratios of D- and L-amino acids or in the assessment of pharmaceutical purity (León-González et al. 2014; Hildmann und Hoffmann 2024; Pirok et al. 2019)."

Line 112, 125 and surrounds. "Both dimensions were measured simultaneously". I think I understand what is going on here but it could be explained more clearly (and perhaps I have misunderstood). If I understand it you are separating a lot of organic compounds in dimension 1, and just cut across to the second dimension for the two target compounds. This allows the enantiomers to appear within the chromatogram at times determined by their retention on both columns, embedded within other compounds whose timing depends only on their retention on one column. I'm not sure I see this as measuring both dimensions simultaneously, rather you ae embedding the results of the second dimension for 2 compounds within a mainly 1D separation. If I have indeed understood correctly, please give a clearer explanation of this.

The reviewer is right in their assumption and we thank them for pointing that this phrasing might be unclear or misleading. We will rephrase our explanation to be more precise:

Line 121: "Both pathways were combined prior to introduction into the mass spectrometer (Fig. 1), allowing for what can be described as simultaneous measurement of the first dimension's full chromatogram alongside the targeted separation of the enantiomers in the second dimension. The targeted enantiomers thus appear at higher retention times influenced by both columns in an otherwise one dimensional separation. In position A, the sample loop is flushed with the eluents from pump 2 (second dimension), while the first dimensional separation is occurring. Upon the elution of the targeted analyte from the first dimension column, the valve is switched to position B, thereby filling the sample loop with the eluent from the first dimension, which contains the analyte peak. The valve is then switched back to position A, allowing the precut volume to be transferred to the chiral column in the second dimension via pump 2 and subsequently to the mass spectrometer."

Fig 3. What is "XIC" and what is the peak at 3 minutes?

We thank the author for pointing this out, "XIC" stands for "Extracted ion chromatogram" as mentioned in the description of Figure 2. We now added the full term in Figure 3 as well to avoid confusion of the reader.

We assume that the reviewer is referring to the peak observed at three minutes in Figure 3a, which belongs to the standard solution (black), as it is the only pronounced peak at this retention time. This peak comprises the remaining unresolved enantiomers of pinic acid in the first dimension. Only a short fraction of this peak was transferred to the second dimension, where the fully separated enantiomers E1 and E2 appear at higher retention times (as labelled in Figure 3a). This also applies to the Belukha sample (red).

Conclusions. It would add to the paper if you discussed at the end what your results imply about the application. I see two points. Firstly with the detection limit around the 1 ppb level, this method is appropriate for mid-latitude cores near to forests but it's worth mentioning it is probably not yet applicable to polar ice cores with lower concentrations. Secondly, as I understand it you are hinting that the measurement of the enantiomers might allow discrimination of particular sources (eg tree species/forest type), or climate conditions. However it seems to me that the fact that you report that the enantiomeric ratios are different for two different oxidation products suggests that the ratio of the sources is not preserved, as does the observation that the predominance of the (-) enantiomer is not what you'd expect for a boreal forest. This is disappointing, and I think it would be appropriate to admit that this makes it difficult to see how the enantiomeric ratios can be used to differentiate sources, even if it suggests other lines of research. In other words, while this paper is an excellent analytical achievement, I think it should admit that its application looks very difficult indeed.

We appreciate the reviewer's comments regarding the need for further investigations, particularly concerning oxidation potential and pathways in the atmosphere that may influence the composition of enantiomeric distributions and ratios. We acknowledge that conducting chamber experiments with enantiomerically pure biogenic volatile organic compounds (bVOCs) and mixtures is essential for better estimations. We understand that this complexity complicates the interpretation of our data in relation to vegetation types and climatic conditions. However, we remain confident in our interpretation that distinguishing chances in vegetation types/climatic changes may be achievable in future studies, provided that these oxidation conditions are adequately considered.

It is important to note that the application of our method to the Belukha ice core samples presented in this study serves primarily as a proof of principle. We view this as a step toward more comprehensive investigations in the future.